# Grafting of Cyclodextrin to Theranostic Nanoparticles Improves Blood-Brain Barrier Model Crossing

**DOI:** 10.3390/biom13030573

**Published:** 2023-03-22

**Authors:** Antonino Puglisi, Noemi Bognanni, Graziella Vecchio, Ece Bayir, Peter van Oostrum, Dawn Shepherd, Frances Platt, Erik Reimhult

**Affiliations:** 1Department of Bionanosciences, Institute of Biologically Inspired Materials, University of Natural Resources and Life Sciences (BOKU), 1190 Vienna, Austria; 2Dipartimento di Scienze Chimiche, Università degli Studi di Catania, 95125 Catania, Italy; 3Central Research Testing and Analysis Laboratory Research and Application Center, Ege University Bornova, Izmir 35100, Turkey; 4Department of Pharmacology, University of Oxford, Oxford OX1 3QT, UK

**Keywords:** cyclodextrins, cholesterol, SPIONs, nanoparticles

## Abstract

Core–shell superparamagnetic iron oxide nanoparticles hold great promise as a theranostic platform in biological systems. Herein, we report the biological effect of multifunctional cyclodextrin-appended SPIONs (CySPION) in mutant Npc1-deficient CHO cells compared to their wild type counterparts. CySPIONs show negligible cytotoxicity while they are strongly endocytosed and localized in the lysosomal compartment. Through their bespoke pH-sensitive chemistry, these nanoparticles release appended monomeric cyclodextrins to mobilize over-accumulated cholesterol and eject it outside the cells. CySPIONs show a high rate of transport across blood–brain barrier models, indicating their promise as a therapeutic approach for cholesterol-impaired diseases affecting the brain.

## 1. Introduction

Superparamagnetic iron oxide nanoparticles (SPIONs) are currently one of the hottest topics in nanomedicine because of their biocompatibility and ease of functionalization, as well as their potential applications in both cancer therapy [1] and imaging [2]. SPIONs are excellent candidates for both targeting and conjugation with biotherapeutics, as they can be retained within tissues for relatively long times and still undergo biocompatible degradation [3,4].

SPIONs first entered clinical use as contrast agents for magnetic resonance imaging [5]. This first generation of clinically used SPIONs consisted of multiple iron oxide cores enwrapped in a polymer matrix. It is a common understanding that a rational design of SPIONs along the lines of core–shell nanoparticles is required to address the goals of theranostic nanomaterials [5,6]. One such novel application is the decoration of SPIONs with therapeutic agents [7] to achieve novel therapeutic effects, slowing clearance, improving targeted biodistribution, and being able to image the distribution of the drug by MRI using the high particle core contrast.

Cholesterol is a major component of cell membranes and plays an essential role in ordinary neuronal physiology [8]. A range of diseases with a very high social impact is closely linked to alterations of cholesterol metabolism, such as cardiovascular and neurodegenerative diseases. The brain contains around 23% of all cholesterol in the body [9]. Defects in its metabolism are linked to severe neurological syndromes, such as Alzheimer’s disease, Huntington’s disease, and Parkinson’s disease, as well as to several rare hereditary diseases including Niemann-Pick disease type C (NPC) [10]. Innovative pharmacological approaches are currently being investigated to counter cholesterol imbalance, particularly in the brain. In this context, the usage of cyclodextrins (CDs) and their derivatives represents a promising therapeutic intervention based on CD’s cholesterol-extracting action [11]. Cyclodextrins are a naturally-occurring family of cyclic oligosaccharides that have been used extensively worldwide in numerous applications in food, cosmetics, agriculture, environment, and pharmaceutics [12,13].

NPC represents a prime example of a disease that may be treated by CD-based therapeutics targeting cholesterol [14]. NPC is an autosomal recessive, prematurely fatal, and rare genetic disorder caused by the accumulation of unesterified cholesterol in the brain, liver, and spleen. In the European Union, NPC has an incidence of around 1 per 100,000. It is caused by mutations in either the Npc1 or Npc2 genes. Most cases are detected during childhood, and progress to cause life-threatening complications by the second or third decade of life. In NPC, the regulation of cholesterol trafficking in the cell is compromised, as the functional role of the proteins encoded by the mutant genes, Npc1 or Npc2, is affected. As a result, an excessive amount of unesterified cholesterol accumulates in late endosomes/lysosomes. The build-up of cholesterol and sphingolipids causes neurodegeneration with subsequent cognitive and mobility decline. Currently, only miglustat is approved as a disease-modifying therapy that significantly slows disease progression [15]. The therapeutic use of CDs is, until now, the most effective approach to stop the progression of the disease in animal models [16], and clinical trials have been or are currently being conducted (NCT01747135, NCT02534844, ClinicalTrials.gov, accessed on 21 March 2023).

The exact mechanism of action of CDs in removing cholesterol is still not fully understood. It may act by extracting cholesterol from sites where it is highly concentrated; that is, in the endosomal–lysosomal compartment, and mobilizing it to the plasma [17,18]. Alternatively, it signals plasma membrane damage and, as a consequence, stimulates lysosomal exocytosis [19,20].

Since the first report in 2008 on using CDs to treat NPC mice via subcutaneous injection [21], this therapeutic approach has seen some remarkable and exciting developments. Clinical studies indicate that a particular chemical modification of CD, the 2-hydroxypropyl-β-cyclodextrin (2HPCD), trademarked as Trappsol^®^ by CTD Holdings, Inc. (now Cyclo Therapeutics Inc., Gainesville, FL, USA) and as VTS-270 by Vtesse, produces beneficial effects in patients with this rare disease.

Although promising, this CD-based treatment for NPC has significant shortcomings, mainly due to poor pharmacokinetics and bioavailability, particularly in the brain. CDs do not cross the blood–brain barrier (BBB) effectively [22]. Systemically administered CDs have a short half-life in the bloodstream owing to their rapid renal clearance [23]. Hence, in order to obtain a sufficient therapeutic effect, high local concentrations (and therefore high doses) of CDs are typically required (2–4 g/kg in mice) [16]. CDs are generally regarded as safe (GRAS) by the FDA. Still, when the dose is high, there is a concern that they may cause toxic effects such as hemolysis, cytotoxicity, ototoxicity, apoptosis induction, and tissue injury [24]. Therefore, the therapeutic efficacy of CDs in their monomeric form is potentially limited due to the combination of poor bioavailability and toxicity at high concentrations [25]. Currently, CD-based therapy for treating the brain in NPC patients via intrathecal administration is underway to obviate these limitations [26].

SPIONs offer additional advantages over polymer nanoparticles in terms of stability and surface modification. Novel synthesis methods yield small superparamagnetic nanoparticles that, including functionalization, can have a hydrodynamic diameter as small as 30 nm [27,28], and could thereby more easily cross the BBB. Indeed, upon proper modification with bioactive coatings, magnetic nanoparticles such as SPIONs have shown penetration across the BBB [29].

We have introduced and described the synthesis of CD-appended SPIONs (CySPION) as cholesterol-mopping agents, specifically designed to release the CD macrocycle from the nanoparticle’s surface at the slightly acidic pH found in the lysosome [30]. These nanoparticles could mobilize cholesterol out of the lysosome to the cytosol and beyond, because of cyclodextrin’s ability to form an inclusion complex with cholesterol.

Here, we investigate CySPION’s biological activity in vitro. Their suitability as a therapeutic platform to remove cholesterol from the lysosomal compartment compared to monomeric cyclodextrins is determined using flow cytometry, while the cellular uptake and localization is investigated using confocal fluorescence and transmission electron microscopy. Moreover, we studied CySPION’s ability to cross the BBB in model systems.

## 2. Materials and Methods

All chemicals, unless otherwise specified, were purchased from Sigma-Aldrich (Hamburg, Germany) and used as received without further purification. 2-methyl-2-oxazoline was dried over CaH_2_ and distilled before use.

Synthesis and Characterization of Nanoparticles

CD-appended nanoparticles (CySPIONs) were obtained according to our previously published protocol [30], and their fluorescently labelled equivalent followed an adapted synthetic strategy, as described.

The fluorescently functionalized PMOXA was obtained through a multi-step, modular synthesis described in the general synthetic scheme (Appendix A).

Hence, a linear poly(2-methyl-2-oxazoline) bearing carboxylic and amine terminations at its two ends, respectively, (J1-en) was synthesized via CROP polymerization [31] under a dry nitrogen atmosphere. 6-bromohexanoic acid (6BHA) 100 mg (0.513 mmol) was used as the initiator and reacted at 110 °C with 8 mL of 2-methyl-oxazoline (94.5 mmol; ×184) in 15 mL of anhydrous dimethylacetamide (DMA) for 22 h. The reaction mixture was then brought to 80 °C and reacted with 1 mL of ethylenediamine (×30 excess) for 22 h to terminate the reaction [32]. After this time, the solution was cooled to room temperature, and the polymer precipitated twice in diethyl ether (200 mL). Finally, the polymer was dialyzed (cut off: 3.5 kDa) overnight and lyophilized to yield ~5 g of J1-en.

^1^*H* NMR (CDCl_3_): δ = 1.00–2.00 (10*H*, 6BHA), 2.10 (282*H*, -*CH_3_CO*-), 3.45 (387*H*, -*OC*-*N-CH_2_-CH_2_-N-* and ethylenic chain of en).

NMR integration (Appendix A) yielded a calculation of ~95 repeating units for the polymer, which is ~8000 Da in molecular weight.

GPC: Mn 8890, Mw 14,467, Mw/Mn 1.6 (Appendix A).

The fluorescently functionalized poly(2-methyl-2-oxazoline) (J1-FITC) was obtained by reacting 2 g of J1-en (0.25 mmol) with 100 mg of fluorescein isothiocyanate (0.25 mmol) in the presence of 36 µL of TEA in 10 mL of DMA 100 °C overnight. The polymer was then dialyzed (cut off: 3.5 kDa) to remove DMA and the excess of fluorescein to obtain 1.52 g of J1-FITC.

NMR reveals the introduction of FITC to the polymer backbone. From integral ratios of the signals of fluorescein and methyls at 2.00 ppm, we determined about 70% of functionalization of the polymer with FITC (Appendix A).

^1^*H* NMR (300 in DMSO): δ (ppm): 1.00–1.64 (CH_2_ of hexanoic acid moiety), 1.80–1.84 (CH_2_ in alpha COOH of hexanoic acid), 2.00 (-*CH_3_CO*-), 3.21–3.55 (-*N*-*CH_2_-CH_2_-N*-, and ethylenic chain of en), 6.40–8.10 (fluorescein H, NHCS), 10.09 (broad band, OH).

Finally, a 6-nitrodopamine (NDA) anchor [33] was introduced into the PMOXA backbone through amide coupling to obtain FITC-PMOXA. 1 g of the carboxy-terminated J1-FITC (0.125 mmol) by dissolving in 12 mL anhydrous DMA in an inert atmosphere. Thereafter, 55 mg (0.145 mmol) TBTU and 22 μL DIPEA (0.126 mmol) were added and stirred for 15 min. NDA (35 mg, 0.125 mmol) was added as a solution in anhydrous DMA. The reaction solution was stirred in the dark for 24 h. The polymer was dialyzed (cut off: 3.5 kDa) against water to remove DMF, the excess of NDA, and the coupling agents to yield ~840 mg of FITC-PMOXA. NMR reveals the introduction of NDA to the polymer. From integral ratios of the signals of the amide signal at 5.7 ppm and *CH*_3_ at 2.00 ppm, we determined a 70% functionalization of the polymer with NDA (Appendix A).

^1^*H* NMR (300 in DMSO): δ (ppm): 1.00–1.64 (CH_2_ of hexanoic acid moiety), 1.80–1.84 (CH_2_ in alpha COOH of hexanoic acid), 2.00 (-*CH*_3_*CO*-), 3.21–3.55 (-*N-CH*_2_*-CH*_2_*-N*- and ethylenic chain of en), 5.80 (CONH), 6.40–8.10 (fluorescein, NHCS, and NDA), 10.09 (broad band, OH).

Core–shell fluorescent nanoparticles (FITC-CySPION) were prepared via ligand exchange using a blend of CD-PMOXA (85% in weight) and FITC-PMOXA (15% in weight) yielding monodisperse, colloidally stable nanoparticles.

100 mg of wet oleic acid-coated SPIONs (from EtOH washing) (Appendix A) were dispersed in 10 mL of DMF together with 500 mg of CD-terminated PMOXA, representing ~17-fold excess with respect to the grafting density of 1 NDA-terminated polymer/nm^2^ and 92 mg of FITC-PMOXA. The dispersion was sonicated for 5 min at room temperature, and the mixture was shaken at 4 °C for 6 days. After this time, the DMF suspension was precipitated with 25 mL of Et_2_O and centrifuged at 4000 rpm for 5 min. The supernatant was discarded, and the residue was washed twice with Et_2_O to remove residual DMF and free oleic acid, leaving a sticky precipitate containing the functionalized nanoparticles and the excess of free polymer.

The core–shell nanoparticles were then purified by dispersing the obtained solid residue in DI water and removing the excess of free polymer using centrifugal filters (Amicon^®^ Sigma-Aldrich, Ultra-15 Centrifugal Filters, RC 30kDa MWCO) at 3000 rpm for 15 min. The operation was repeated several times until the separated solution appeared free of polymer.

*NMR* spectra were collected on a Bruker Avance III HD 300 MHz and processed with Bruker Topspin 3.5 PL six software.

*Dynamic light scattering (DLS)* was used to determine the size of FITC-CySPION. A solution of the fluorescent nanoparticles in PBS buffer at 0.3 mg/mL revealed an average hydrodynamic diameter of 240 nm (PDI 0.232) (Appendix A). This was considerably bigger than the diameter found for the unlabeled CySPION, which was found to be 77 nm (PDI 0.217).

*TEM* studies were performed on an FEI Tecnai G2 20 transmission electron microscope operating at 120 or 200 kV for high-resolution imaging. Samples were prepared by dropping toluene dispersions of oleic acid-coated iron oxide core nanoparticles onto a 300-mesh carbon-coated copper grid and subsequently evaporating the solvent in air. TEM thin section was used to visualize the uptake of CySPIONs with CHO cells. Embedding of cells was undertaken in LR-White acrylic resin, according to a modified protocol of Glauert and Lewis [34]. Briefly, after uptake, cells were washed twice in 0.1 M sodium cacodylate pH 7.4 and fixed in fixative containing 2.5% glutardialdehyde, 2.5% paraformaldehyde, 2.5 mM CaCl_2_, and 1% tannic acid in 0.1 M sodium cacodylate pH 7.4 for 4 h. Fixation was repeated with fixative without tannic acid for 20 h at 4 °C. After washing with sodium cacodylate followed by distilled water, cells were postfixed with 1% OsO_4_, 1.5% potassium hexacyanoferrate (III) in water for 1 h, followed by 2% OsO_4_ in water for an additional 2 h at room temperature. After brief washing in water, cells were dehydrated using a graded ethanol series in water (70–80–90%–2 × 100%) for 10 min each. Cells were infiltrated with LR-White for 30 min, followed by an additional incubation with fresh resin overnight at 4 °C. Samples were transferred into gelatin capsules size 00 and filled with plain resin. Blocks were cured at 60 °C for a minimum of 24 h and stored at room temperature. Ultrathin sections were cut using Leica Ultracut UC-7. 70 nm slices of fixed and embedded cells were transferred onto 150 mesh hexagonal copper grids coated with Pioloform (Wetzlar, Germany). After air drying, samples were investigated without further staining.

*Thermogravimetric Analysis (TGA)* and Differential Scanning Calorimetry (DSC) measurements. Thermograms were recorded on a Mettler-Toledo TGA/DSC 1 STAR system in the temperature range 25–650 °C with a ramp of 10 K/min in a synthetic air stream of 80 mL/s to ensure complete combustion of ligands, as NDA was found to polymerize by pyrolization under N_2_. 70 μL aluminum oxide crucibles were filled with 0.5–1.5 mg of sample, and the total organic content (TOC) was evaluated as the mass loss fraction at 500 °C by horizontal setting. The density of grafted polymer was calculated from the inorganic/organic fraction of purified CySPION (Appendix A), as reported in our previous work [30]. This in turn allowed us to estimate that the equivalent concentration in appended CD, that is, 1 mg/mL CySPION, was equivalent to ~120 μM in monomeric CD.

*Cell Culture*. WT and Npc1-deficient CHO cells were a gift of Frances Platt, Department of Pharmacology, University of Oxford. CHO cells seeded in DMEM containing GlutaMax and HEPES, supplemented with 10% fetal calf serum and 100 units/mL of penicillin/streptomycin, were grown as monolayers at 37 °C with 5% CO_2_.

*Lysotracker flow cytometry*. Npc1-deficient CHO lines were grown in DMEM/F12 TC media (Gibco: 31330-038) supplemented with 10% FBS, glutamine, and pen/strep. An amount of 20,000 CHO NPC−/− (null) cells or WT CHO were plated in 12 well TC plates and allowed to adhere. Media was then replaced with CySPION, CDen, or 2HPCD at different concentrations, and the plates were incubated at 37 °C, 5% CO_2_ for 72 h.

*Lysotracker staining*. LysoTracker staining was performed, as described previously [35]. In brief, cells were washed twice in situ with PBS and harvested using Trypsin/EDTA. Cells were washed again with PBS and stained with 200 nM LysoTracker-green DND-26 (Thermo Fisher, Bremen, Germany) for 10 min in the dark. Cells were washed a final time with PBS and re-suspended in a buffer containing 5 μg/mL propidium iodide (Sigma) to allow for the exclusion of dead cells, and immediately analyzed on a BD FACS-Canto II (Beckton Dickinson, Wokingham, UK). A minimum of 10,000 events were collected for each sample, and relative fluorescence values were calculated using FlowJo software (Version 10, FlowJo, LLC, Ashland, OR, USA).

*Confocal microscopy*. FITC-CySPION uptake in Npc1-deficient CHOs and the co-localization in the lysosomes was determined by means of confocal laser scanning microscopy (CLSM) using an SP8 by Leica. The DNA label Hoechst 34,580 was excited with the 405 nm diode laser, while the fluorescein in the nanoparticle and the Lysotracker Deep Red was excited with the 495 nm and 653 nm lines of the microscope’s white laser. To minimize the crosstalk between the signal from the DNA label and the nanoparticle, mainly caused by the 405 nm laser exciting the fluorescein as well, we applied line-based sequential scanning, where the DNA (380 nm–385 nm) and the Lysotracker (658 nm–776 nm) were first imaged and the CySPIONs (500 nm–617 nm) was subsequently imaged. During the second exposure, we also collected the transmitted 495 nm light to produce a transmission micrograph.

*Co-cultured BBB model*. An in vitro BBB model was created by cultivating a human brain microvascular endothelial cell line (HBEC-5i, CRL-3245TM, ATCC^®^, Manassas, VA, USA) and a mouse brain astrocyte cell line (C8-D1A, CRL-2541, ATCC^®^, Manassas, VA USA). The cell cultures were incubated in a 37 °C, 5% CO_2_ and 95–98% humidified incubator (SteriCycle 160i, Thermo Scientific, Waltham, MA, USA). Dulbecco’s Modified Eagle’s Medium F12 (DMEM-F12, D6421, Sigma-Aldrich), containing 40 μg/mL of endothelial cell growth supplement (ECGs, E2759, Sigma-Aldrich), 10% fetal bovine serum (FBS, 16000044, Gibco, Thermo Scientific), and DMEM (D6046, Sigma-Aldrich) media supplemented with 10% FBS were used for the cultivation of endothelial and astrocyte cells, respectively. The culture media were changed every other day, and the cells were passaged at a 1:3 split ratio until desired cell numbers were obtained. Cell culture inserts (353095, Falcon^®^ Corning, Sigma-Aldrich) were conditioned in endothelial growth medium for 2 h. Inserts were turned upside down, and astrocytes were seeded onto the underside of the insert membrane at 5 × 10^5^ cell/cm^2^ concentration. After the astrocyte cells were incubated for 4 h, the inserts were turned, and endothelial cells were seeded on the apical surface of the membrane at 1 × 10^6^ cell/cm^2^ concentration. Obtained BBB models were incubated for 5 days, and LY permeability was determined to characterize the model and validate its integrity (see below).

*Permeability study*. The above-described cocultured BBB model was used to test the permeability of CySPION and free polymer samples. First, BBB integrity was assessed via LY permeability analysis. Ringer’s HEPES solution (150 mM NaCl, 3.4 mM CaCl_2_, 1.2 mM MgCl_2_, 5.2 mM KCl, 0.5 mM NaHCO3, 2.8 mM glucose, and 10 mM HEPES) was prepared in type-I ultra-pure water. Lucifer Yellow CH dipotassium salt (LY, L0144, Sigma-Aldrich) was dissolved in Ringer’s HEPES solution (1 mM). After rinsing the BBB model with Ca^2+^ and Mg^2+^ free phosphate-buffered saline (PBS), LY solution was added to the apical part, and Ringer’s HEPES solution was added to the basolateral part of the model. LY solution was transported through the models for 30 min, and RFU values of the solution in the basolateral part were measured using a fluorospectrometer (Nano Drop 3300, Thermo Scientific) at 530 nm. Concentration values corresponding to the obtained RFU values were determined using the calibration graph (R^2^ = 0.9997) of the LY solution (Appendix A). Permeability values were calculated according to the formulae below (where P is the permeability, V is the volume of the media in the basolateral part, A is the surface area of the insert membrane, and [C] is the concentration of the LY).
$$P\left(\frac{cm}{s}\right)=\frac{V\left(cm^{3}\right)}{A\left(cm^{2}\right)\times {\left[C\right]}_{apical}\left(\frac{g}{mL}\right)}\times \frac{\Delta {\left[C\right]}_{basolateral}\left(\frac{g}{mL}\right)}{\Delta t\left(s\right)}$$

A permeability value < 2 × 10^−^^6^ cm/s for LY indicates the establishment of a good cell barrier layer [36]. The model used in this permeability study had a lower LY permeability value than models in the literature [37]. The optimal permeability value of LY was obtained at 30 min (1.39 × 10^−^^6^ ± 4.41 × 10^−^^7^ cm/s) in this study, which was chosen as a suitable assay time. The test samples were prepared in Ringer’s HEPES solution at 1 mg/mL concentration. Calibration graphs were obtained for both FITC-PMOXA (R^2^ = 0.9997) and FITC-CySPION (R^2^ = 0.9902) for the relationship between concentration and RFU values at 520 nm using fluorospectrometery, with the fluorescence provided by the FITC tag attached to both FITC-PMOXA and FITC-CySPION (Appendix A for calibration curve of FITC-PMOXA and S13 for calibration curve of FITC-CySPION in Appendix A). The permeabilities for FITC-PMOXA and FITC-CySPION, respectively, were calculated using the same formula as for the LY assay, using these calibrations.

## 3. Results and Discussion

The design strategy underpinning the synthesis of the CySPIONs has already been described in detail by our group [30], building on previous work on polyalkyloxazoline-grafted SPIONs for controlled cell uptake [38]. Briefly, by using a highly functional nitrodopamide-terminated poly(2-methyl-2-oxazoline) (PMOXA) polymer bearing the CD macrocycle through a pH-cleavable linkage, we were able to obtain well-defined core–shell nanoparticles that release CDs at lysosomal pH. In this study, we investigated their cytotoxicity profile, in vitro activity (i.e., intracellular cholesterol mopping activity), and cellular uptake in an Npc1-deficient Chinese Hamster Ovary (CHO). We chose this cell line because of its mutations in the Npc1 gene [39], leading to defects in intracellular cholesterol metabolism and subsequent accumulation [40,41].

CySPIONs did not show any significant cytotoxicity compared to a control in the resazurin viability assay on CHO cells within the explored concentration range (up to 1 mg/mL) (Appendix A).

The desired therapeutic effect of CySPIONs is to remove the excess cholesterol in cholesterol-impaired diseases. Hence, CySPIONs’ cholesterol-solubilizing activity on Npc1-deficient CHO was assessed via a Cholesterol Assay Kit (Sigma-Aldrich MAK043) [42]. The solubilizing activity was compared to the capacity of the same monomeric cyclodextrin used for the synthesis of the nanoparticle (ethyl diamino-β-cyclodextrin: CDen), at the same concentration as estimated for the CDs appended to the CySPION (Figure 1). We assessed the average number of CDs per CySPION via TGA. 1 mg/mL of CySPION was previously found to be equivalent to 120 µM CD [30]. Free monomeric CD removes a significant amount of cholesterol from Npc1-deficient CHO at 0.5 mg/mL. We observed a similarly significant amount removed by CySPIONs at twice the equivalent CD concentration. Although the quantification of CD grafted to the CySPION is not precise, these results suggest that the accessibility of cholesterol to the CD on the CySPION, or their uptake and release, is slightly less efficient than those of free monomeric CD. This could tentatively be remedied by optimizing the assay to account for both uptake and cleavage of CD from the CySPION, as only cholesterol mobilized by cleaved-off CD leaving the cell will be quantified by the assay. In contrast, monomeric βCD could also remove cholesterol from the outer cell membrane and make it directly detectable in the supernatant, hence making it active on a shorter time scale.

A potential advantage of using nanoparticles to administer CD is the expected higher endosomatic internalization of nanoparticles compared to free monomeric CD. Continuing from the endosome, a nanoparticle-releasing CD could sequester cholesterol directly from the lysosome, where it is accumulated in cells with cholesterol enrichment pathologies. Hence, we were interested in demonstrating that CySPIONs are predominantly taken up into the endosome/lysosome and quantifying their effect on the CHOs’ enlarged lysosomes.

We measured the relative volume of the lysosomal compartment in controls, without and in the presence of CDen or CySPION using the LysoTracker assay and flow cytometry [35]. LysoTracker probes are weakly basic amines that selectively build up in the lysosomal acidic compartment. Their fluorescence is proportional to the total volume of these compartments within the cell. It is generally assumed that the higher fluorescence of mutant Npc1-deficient CHO cells compared to wild type (WT) CHO is due to larger individual lysosomal vesicles, swollen by the high amounts of cholesterol and sphingolipids that are accumulating in them. We expected the LysoTracker signal to go down on average if cholesterol was extracted from the lysosome by CDen or CySPIONs, making the lysosomal volume of the cell smaller.

LysoTracker assay results showed a modest but significant decrease in lysosomal volume when CHOs were treated with CySPIONs (Figure 2). In line with the cholesterol mopping activity measurements, the lysosome volume decrease was smaller than in the control with an equivalent concentration of free monomeric CDs, with 2HPCD performing much better than CDen. At a low (0.12 mg/mL) concentration, the reduction in lysosome size is equal between CySPION and the equivalent amount of free monomeric CDen. While the lysosomal size reduction is increased by increasing the concentration of monomeric CD, the lysosome size remains near-constant as the CySPION concentration is increased to 1 mg/mL.

We hypothesized that in comparison to the free CD, the larger volume occupied by the CySPIONs in the lysosome, combined with the kinetics of lysosome uptake and subsequent CD cleavage from the CySPION shell, leads to a short-term saturation of activity. However, this could potentially lead to a more extended effect of keeping the cholesterol concentration down over time. Supporting this hypothesis is that the side scatter intensity, measured in the flow cytometry analysis (Figure 3A,B), increased in the presence of the nanoparticles. A more complex, granular intracellular architecture is the likely cause of the increased side scattering intensity, and could be explained by the uptake of nanoparticle clusters into the lysosome.

It should be noted that Npc1-deficient CHO cells are more granular (increased side scatter) than WT cells (Figure 3B). This is due to the build-up of cholesterol-rich, multi-concentric, lamella-like structures in the perinuclear vesicles of these cells [41]. Treatment with soluble cyclodextrin decreases granularity, presumably by reducing the size/number of these structures. CySPION treatment increases cell granularity due to the accumulation of nanoparticles inside the cell and/or on its surface. Due to both CDen and CySPION treatment reducing the LysoTracker signal, we can postulate that the lamella-like structures are also reduced in CySPION-treated cells, but that the presence of the nanoparticles masks this effect.

Transmission electron microscopy (TEM) thin section images of fixed and embedded Npc1-deficient CHO cells exposed to CySPIONs, further supported the hypothesis that CySPIONs are internalized into the cell. They show nanoparticle clusters both in the endosome/lysosome and associated with the cell surfaces (Figure 4). The clustering of the CySPIONs for uptake creates a very high local concentration of CD, and may slow down both uptake and release.

We additionally prepared fluorescent CySPIONs (FITC-CySPION) by grafting a minor fraction (15%) of fluorescently-functionalized PMOXA (FITC-PMOXA) together with the CD-functionalized PMOXA to the iron oxide core, giving us the ability to track the nanoparticles’ fate within the cells and assess their biological activity. The successful preparation and characterization of highly fluorescent FITC-CySPION are described in the Experimental Section and Appendix A.

Npc1-deficient CHOs were exposed to the fluorescently labeled FITC-CySPIONs to follow their uptake and locate them in the cells. The cells were labelled with the DNA stain Hoechst 34,580 and LysoTracker immediately prior to confocal inspection. Figure 5 shows confocal fluorescence images of the cell nuclei (blue), the green-emitting FITC-CySPIONs, and the LysoTracker stain for the lysosome (red), as well as an overlay of these channels. These images show the location of the lysosomes close to the nuclei, and a strong co-localization of FITC-CySPION and lysosomes with a Pearson’s colocalization coefficient of 0.37, indicating an intermediate colocalization. There are many CySPION stuck to the glass bottom outside the cells, which significantly reduces this quantitative measure. There is little indication of free FITC-CySPION in the cytosol and nanoparticles attached to the cell surface. We conclude that they are predominantly endocytosed to enter the cells, as expected for nanoparticles of this size [43].

Finally, we investigated CySPIONs’ ability to cross the BBB, as this seems to be one of the main obstacles for CDs in the treatment of neurodegenerative diseases. The permeability of SPIONs through an in vitro BBB model was already investigated by Shi et al. with promising results [29].

We created an in vitro BBB model by co-cultivating a human brain microvascular endothelial cell line (HBEC-5i) and a mouse brain astrocyte cell line (C8-D1A). A Lucifer Yellow (LY) permeability test was then performed to establish the integrity of the BBB model by measuring that the permeability was in the accepted range (<2 × 10^−^^6^ cm/s after 30 min).

To assess the effectiveness of the nanoparticles in promoting BBB crossing of the CDen, we compared FITC-CySPION with an equivalent amount in mass and composition of the blend of free polymers (85% CD-PMOXA + 15% FITC-PMOXA) used to make such nanoparticles. The permeabilities of the blend of free polymers and FITC-CySPION, respectively, were then measured for the BBB model by determining their concentrations transported across the layer from the fluorescence signals of FITC-PMOXA and FITC-CySPION, respectively. Briefly, FITC-PMOXA or FITC-CySPION was seeded on the apical side of the BBB model, while a Ringer’s HEPES solution was added to the basolateral side. The fluorescence (in RFU values) of the test samples transported through the barrier was then measured after 30 min, and the permeability value was calculated using calibration curves for the respective samples (Appendix A).

It was found that the permeabilities of free polymer and FITC-CySPION were 1.25 × 10^−^^6^ ± 6.89 × 10^−^^7^ cm/s, and 3.78 × 10^−^^6^ ± 8.27 × 10^−^^7^ cm/s, respectively (Figure 6), indicating approximately three times higher BBB crossing activity for the nanoparticle system in comparison to the free polymer. This is in line with the literature for similar systems [44].

## 4. Conclusions

In summary, we have demonstrated that CD-appended SPIONs, CySPIONs, provide a controlled way of simultaneously incorporating multiple functions that could benefit a drug delivery system targeting NPC or other diseases related to cholesterol imbalances in the brain. The multifunctional nanoparticles show negligible cytotoxicity. They are strongly endocytosed and localized in the lysosome of cells, where the pH-sensitive linker releases the appended CD. At least a similar amount of cholesterol is mobilized and exported outside the cells, as for monomeric CD. There is the potential for further improvement by tailoring its longer retention time, slower uptake and release than CDen. The significantly higher rate of transport of CySPION than free polymer through a model BBB could indicate a decisive advantage of the responsive, nanoparticle-based delivery system. Hence, this nanosystem demonstrates excellent potential as a therapeutic system for cholesterol-impaired diseases, deserving further investigation. Such continued in vivo studies will be facilitated by the built-in dual imaging modalities of fluorescence and magnetic resonance contrast.

## Figures and Tables

**Figure 1 biomolecules-13-00573-f001:**
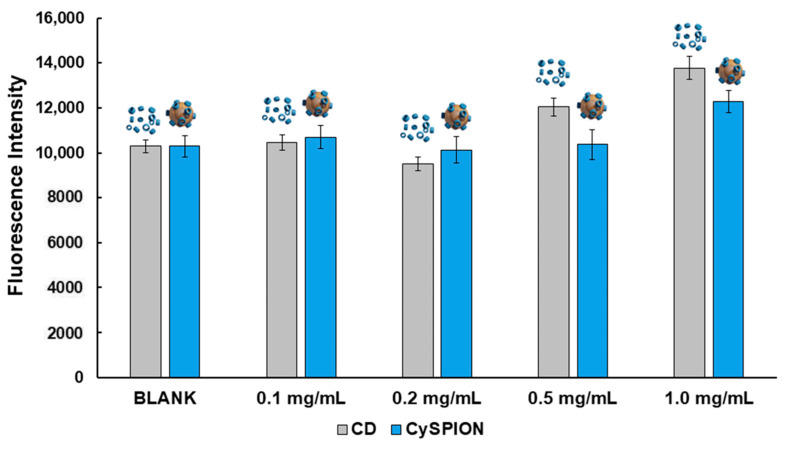
Cholesterol quantification assay using an enzymatic kit in Npc1-deficient CHO cells for CySPION and CDen. The CySPION solubilizing effect on cholesterol was slightly less efficient in comparison to the monomeric macrocycle at an equivalent CD concentration.

**Figure 2 biomolecules-13-00573-f002:**
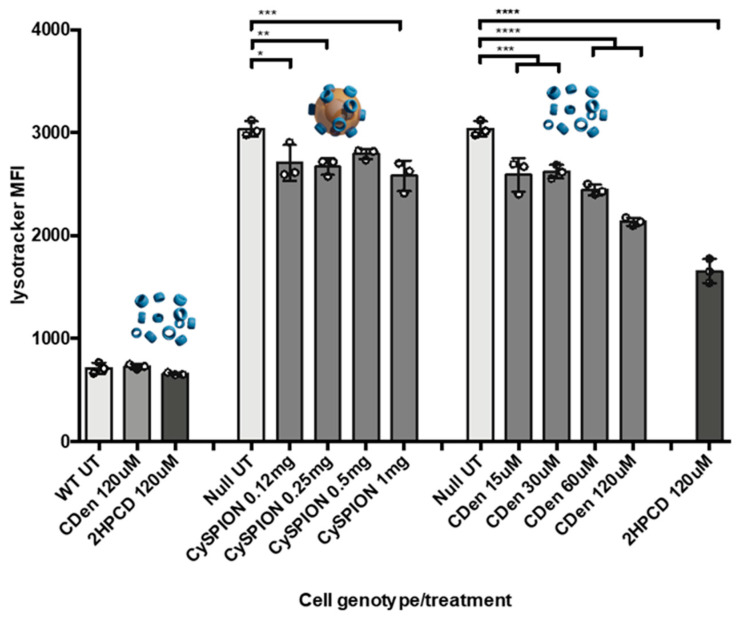
Lysotracker^TM^ staining of CDen treated CHO cells. LysoTracker^TM^ staining of WT and Npc1-deficient CHO cells treated with the indicated doses of CySPION, CDen and 2HPCD for 72 h. 1 mg CySPION dose is equivalent to 120 µM CDen. Data are mean ± SD, *N* = 3 replicates per sample. Statistical analysis, one-way ANOVA, **** *p* < 0.0001, *** *p* < 0.001, ** *p* < 0.01, * *p* < 0.05. Data are representative of two independent experiments.

**Figure 3 biomolecules-13-00573-f003:**
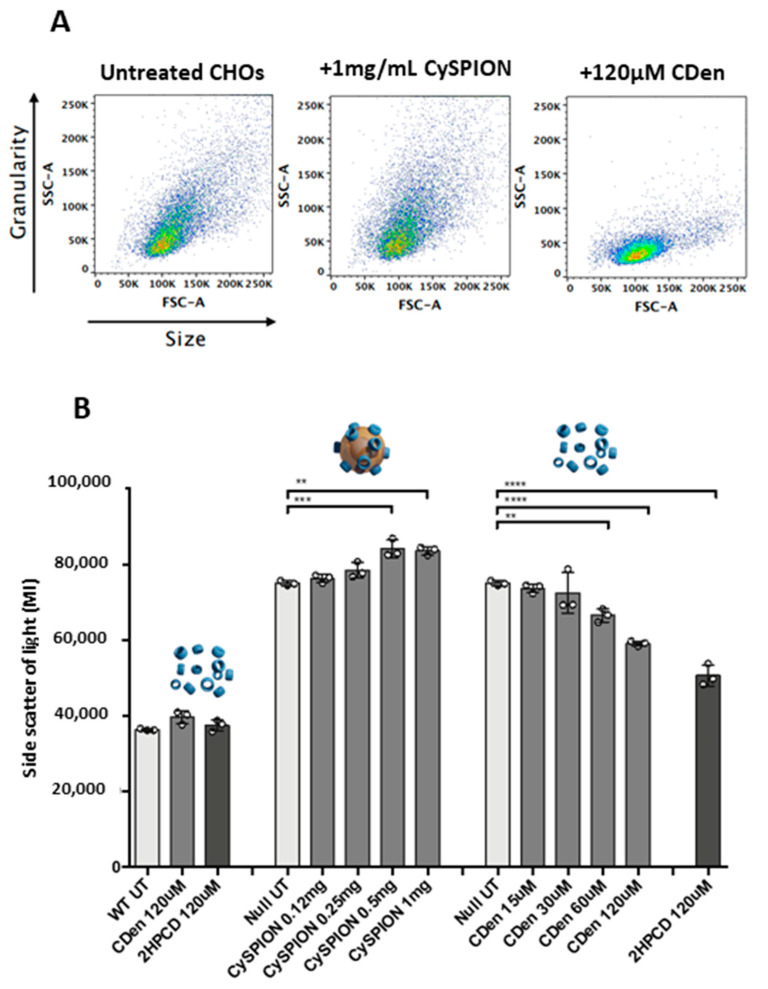
(**A**) Npc1-deficient CHO Cell profile on flow cytometer. Side scatter (SSC-A) vs. forward scatter (FSC-A). (**B**) Side scatter of WT and Npc1-deficient CHO cells treated with the indicated doses of CySPION, CDen, or 2HPCD for 72 h. A 1 mg CySPION dose is equivalent to 120 µM CDen. Data are mean ± SD, *N* = 3 replicates per sample. Statistical analysis, one-way ANOVA, **** *p* < 0.0001, *** *p* < 0.001, ** *p* < 0.01. Data are representative of two independent experiments.

**Figure 4 biomolecules-13-00573-f004:**
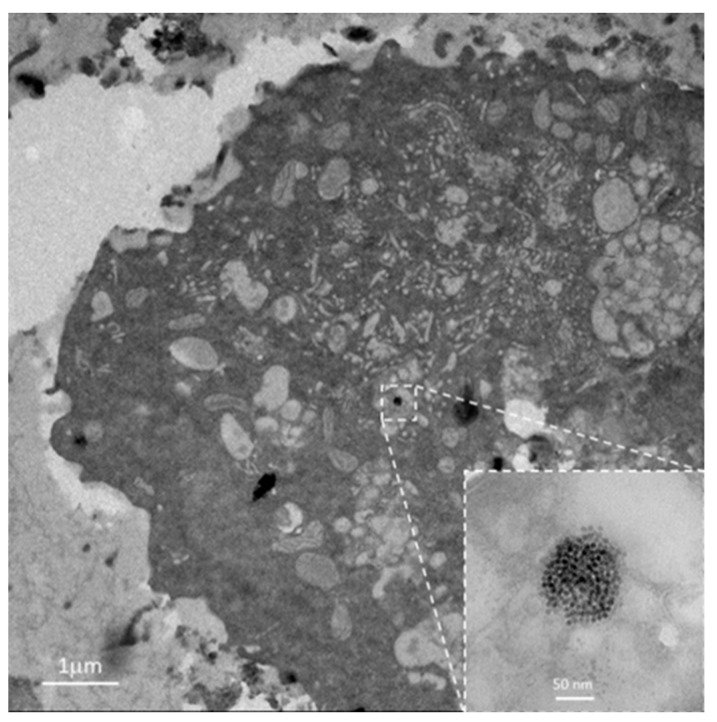
TEM thin section micrographs of fixed and embedded Npc1-deficient CHO exposed to CySPION supporting nanoparticles’ cellular internalization.

**Figure 5 biomolecules-13-00573-f005:**
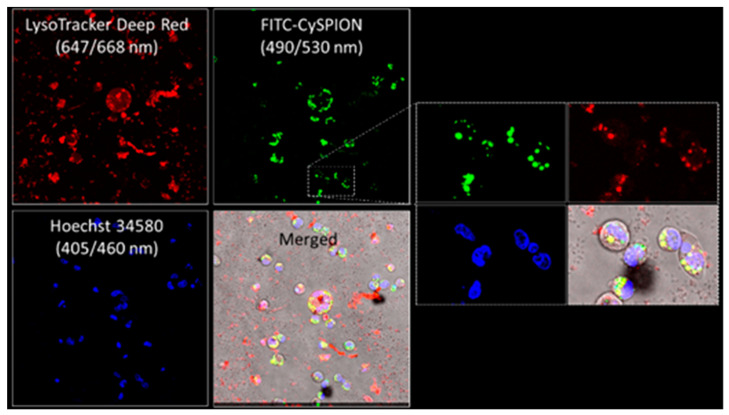
Confocal micrographs of Npc1-deficient CHO after incubation with 0.1 mg/mL FITC-CySPION for 72 h showing co-localization, with Pearson’s coefficient of 0.37, between FITC-CySPION (green) and LysoTracker Deep Red within the lysosomal compartments (red) within the ROI preproduced on the right. The used excitation wavelengths and fluorescence maxima are indicated in the figures. The field of view is 290 × 290 µm^2^.

**Figure 6 biomolecules-13-00573-f006:**
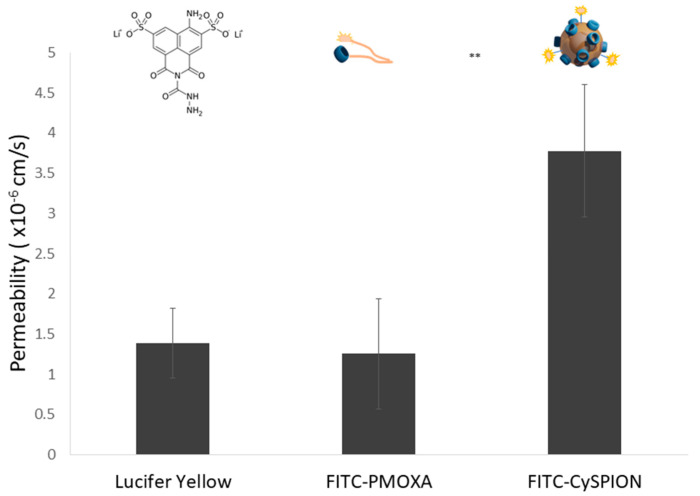
Permeabilities obtained by passing LY, FITC-PMOXA, or FITC-CySPION for 30 min on the fifth day of the Transwell^®^ BBB model obtained by co-cultivating HBEC-5i and C8-D1A cells. It shows a significantly higher permeability of FITC-CySPION. The data were statistically analyzed by unpaired, two-tailed *t*-test method using the GraphPad Prism 8 program (** *p* < 0.005).

## Data Availability

The original contributions presented in the study are included in the article/Appendix A. Further inquiries can be directed to the corresponding author.

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
