# Peer review of "Grafting of Cyclodextrin to Theranostic Nanoparticles Improves Blood-Brain Barrier Model Crossing"

_biomolecules, 2023, doi:10.3390/biom13030573_

Round 1
Reviewer 1 Report
The manuscript is well written and of high impact in the field. I have only a few minor comments.
Figure 1 lack a y axis title. The figures in general are not that well described in the figure text. It is not a requirement, but a TEM in a lower magnification where more of the tissue/cell structures could be identified, would help the reader understand the distribution of the cy-SPIONS.
Author Response
Reviewer's comment
Figure 1 lack a y axis title. The figures in general are not that well described in the figure text. It is not a requirement, but a TEM in a lower magnification where more of the tissue/cell structures could be identified, would help the reader understand the distribution of the cy-SPIONS.
Response
Y-axis in Figure 1 has now been added
Reviewer 2 Report
In this manuscript, the authors have reported biological effect of core-shell superparamagnetic iron oxide nanoparticles (SPIONs) functionalized with cyclodextrin. They have shown that these biologically compatible NPs localizes in the lysosomes and releases cyclodextrins which aid to eject over-accumulated cholesterol outside the cells. Since these NPs can cross Blood-Brain Barrier model, it could potentially be useful for cholesterol impaired diseases affecting the brain.
I would recommend accepting the manuscript after minor revisions.
Minor comments
1. The authors have stated that granular intracellular architecture is the likely cause of the increased side scattering intensity. But only nanoparticle intake could potentially increase side scattering intensity. How do the authors ensure that it is due to granularity?
2. Mention the Y-axis in Figure-1
3. In Figure-5, it has been shown that Lyso tracker and FITC-CySPION are co-localized. It will be good to provide its quantitative estimation such as calculating Pearson’s correlation coefficient.
4. To calculate the permeability, at what pH has the calibration curve been determined? Because FITC is a pH sensitive dye, and the NPs gets accumulated in the lysosomes which is acidic.
5. Why does the size increase (from 77 nm to 240 nm) upon labelling? Does the particle aggregate? Or do the hydrodynamic radii increase only? TEM image can revolve this.
Author Response
Reviewer's comment 1
The authors have stated that granular intracellular architecture is the likely cause of the increased side scattering intensity. But only nanoparticle intake could potentially increase side scattering intensity. How do the authors ensure that it is due to granularity?
Response 1
We are sorry for the confusion. We refer to that particle uptake into the endosome/lysosome increases the scattering contrast of these organelles and in that sense increases the optical granularity of the cells. We cannot directly infer that the increase in side scattering is caused by this effect, but it is consistent with our other observations and with well-documented results from flow cytometry and optics. The change in side scatter of NPC1-/- CHOs versus WT measured on a flow cytometer is proportional to the granularity of the cell as NPC1-/- CHO have higher side-scatter than WT due the storage of cholesterol in lamella-like structures which would also refract the light increasing side-scatter measurements.
Reviewer's comment 2
Mention the Y-axis in Figure-1
Response 2
Y-axis in Figure 1 has been added
Reviewer's comment 3
In Figure-5, it has been shown that Lyso tracker and FITC-CySPION are co-localized. It will be good to provide its quantitative estimation such as calculating Pearson’s correlation coefficient.
Response 3
This point has been now expanded including Perason’s correlation in lines 394-396 of the updated manuscript
Reviewer's comment 4
To calculate the permeability, at what pH has the calibration curve been determined? Because FITC is a pH sensitive dye, and the NPs gets accumulated in the lysosomes which is acidic.
Comment 4
Permeability was calculated by using a calibration curve in PBS. Though FITC is pH sensitive the measured amount of dye has already been transported across the BBB model and is, therefore, no longer at the lysosomal acidic pH.
Reviewer's comment 5
Why does the size increase (from 77 nm to 240 nm) upon labelling? Does the particle aggregate? Or do the hydrodynamic radii increase only? TEM image can revolve this.
Response 5
It is likely that the major part of the increase in the NPs’ hydrodynamic sizes measured after labelling with the imperfectly soluble dye FITC by DLS increases upon in the labelled version because of some is due to minor aggregation including small clusters. H, however there might also be a contribution of extended hydrodynamic radius due to FITC moiety. TEM cannot resolve this as the aggregation state of the particles on the surface of the grid are predominantly determined by the spreading and drying conditions of the particles before transfer to the TEM vacuum. Spreading of the particles on the surface from water invariably leads to the observation of drying effects due to the high surface tension of water.
Reviewer 3 Report
The manuscript involves the study of the cytotoxicity profile, in vitro activity and cellular uptake of SPIONs decorated with cyclodextrin (CySPIONs) with improved transport across BBB models to act as therapeutic platform to remove cholesterol from the lysosomal compartment. The topic is timely and though, as acknowledged in the manuscript, the synthesis of CySPIONs has already been described in previous work of the group, results are quite interesting. In my opinion, the manuscript should be accepted for publication in Biomolecules after minor revision.
1. Line 118 – It would be interesting to discuss how the concentration range (up to 1 mg/ml) explored in the cytotoxicity assays compare to the "expected" required amount of CySPIONs in relation to body weight?
2. Lines 341 to 343 – please also inform the dispersity values (PDI) of labeled and unlabeled CysSPIONs.
3. Lines 446, 447 – please complete information related to the Supplementary Materials (titles of Figures and Tables of SM)
4. Line 233 – use superscript in 10-6. The same in lines 250, 251, 438
5. Supplementary Materials – please verify if in the caption of figure S8 “DSC” should be replaced by “TGA”
Author Response
Reviewer's comment 1
Line 118 – It would be interesting to discuss how the concentration range (up to 1 mg/ml) explored in the cytotoxicity assays compare to the "expected" required amount of CySPIONs in relation to body weight?
Respinse 1
Systemically administered CDs need a high doses (2-4 g/kg in mice) in order to exert sufficient therapeutic effect (Ref 16). However, improved BBB crossing might considerably lower that value. The estimated concentration range for our system in vivo for the moment remains beyond the objective of this study.
Reviewer's comment 2
Lines 341 to 343 – please also inform the dispersity values (PDI) of labeled and unlabeled CysSPIONs.
Response 2
PDI values have been added
Reviewer's comment 3
Lines 446, 447 – please complete information related to the Supplementary Materials (titles of Figures and Tables of SM)
Response 3
Amended
Reviewer's comment 4
Line 233 – use superscript in 10-6. The same in lines 250, 251, 438
Response 4
Amended
Reviewer's comment 5
Supplementary Materials – please verify if in the caption of figure S8 “DSC” should be replaced by “TGA”
Response 5
Yes, in Figure S8 of the Supplementary material should read “TEM and TGA of oleic acid-coated SPIONs obtained via heat-up method”. An amended file for supplementary material has been uploaded.